# Statistical Treatment of Clinical Investigations in Pediatric Cardiology

**DOI:** 10.3390/children8040296

**Published:** 2021-04-12

**Authors:** P. Syamasundar Rao

**Affiliations:** University of Texas-Houston McGovern Medical School, Children’s Memorial Hermann Hospital, Houston, TX 77030, USA; P.Syamasundar.Rao@uth.tmc.edu or srao.patnana@yahoo.com

**Keywords:** normal distribution, mean, median, standard deviation, standard error of mean, paired *t* test, analysis of variance, chi-square tests, Bonferroni correction, multivariate logistic regression, actuarial analysis of event-free rates, linear regression, correlation coefficients, simple and multiple linear regression analysis, contingency tables, inter-observer and intra-observer variation

## Abstract

This paper describes various statistical methods used by the author during multiple studies conducted by the author. Initially, the data were scrutinized to ensure normal distribution, and expressed as mean ± standard deviation (SD) or standard error of mean (SEM) for normally distributed variables. Medians and ranges were given for the data with skewed distribution. Two tailed, paired *t* tests or independent sample *t* tests (analysis of variance) were used for normally distributed data, while non-parametric chi-square or similar other tests were utilized for data with skewed distribution. Statistical significance was set at a *p* value of < 0.05. Bonferroni correction was applied when the study involves multiple comparisons. A number of other statistical methods used during these studies were also discussed. Finally, special methods used in evaluating aortic remodeling subsequent to balloon angioplasty of native aortic coarctation were reviewed.

## 1. Introduction

The author has been involved in a number of clinical investigations over the years. Consequently, it was surmised that knowledge of statistics is of immense importance in the conduct of these studies. The author has utilized a formal course in statistics that he took at Stanford University during the first year of his pediatric cardiology fellowship along with the assistance of several statistics books to expand the author’s statistical horizon. In addition, a number of statisticians - Robert J. Haggard and Rollie J. Harp of the Medical College of Georgia, Patrick Carey, Rebecca Langhough, and Rebecca Koscik of University of Wisconsin, and Brain Waterman and James W. Hormann of St. Louis University—have helped the author in the statistical analysis and interpretation of the data in several clinical investigations [1]. The purpose of this review is to present the approach that the author has utilized in the statistical treatment of clinical investigations in pediatric cardiology with the expectation that this information is useful to the investigators interested in advancing knowledge in pediatric cardiology.

## 2. Standard Statistical Analysis

Initially the data were examined to see if they were normally distributed (Gaussian distribution); this was usually performed using the Kolmogorov-Smirnov method or similar tests. Data were expressed as mean ± standard deviation (SD) or standard error of mean (SEM) for continuous, normally distributed variables. Medians and ranges were given when the data were not normally distributed (i.e., data with skewed distribution). Values before and after an intervention (for example, catheter interventions before and after balloon dilatation, or between the pre-occlusion and post-occlusion of cardiac defects) were compared by two-tailed or paired *t* tests. Independent sample *t* tests (analysis of variance) were used for between-group comparisons of normally distributed variables. Fisher’s exact, Kruskal-Wallis, McNemars, or Mann-Whitney tests or other chi-squared tests were used, as appropriate, for between-group comparisons of categorical, ordinal, or not normally distributed variables. The level of statistical significance was set at *p* < 0.05. When multiple comparisons were made, the Bonferroni correction was applied [2]. The methods of standard statistical analysis used in the studies are listed in Table 1.

## 3. Other Types of Statistical Analysis for Special Circumstances

A number of other types of statistical method were utilized depending upon the type of study. These are listed in Table 2.

### 3.1. Multivariate Logistic Regression

Multivariate logistic regression analysis was performed to identify predictors of restenosis following balloon dilatation for stenotic lesions such as pulmonary stenosis, aortic stenosis and coarctation of the aorta [3,4,5,6,7].

### 3.2. Actuarial Analysis of Event Free Rates

The actuarial analysis of event-free rates was calculated using the Kaplan-Meir method [8]. This analysis was used to determine event-free rates following balloon pulmonary valvuloplasty (Figure 1) [9], balloon angioplasty of aortic coarctation (Figure 2) [10], balloon aortic valvuloplasty (Figure 3) [11], trans-catheter occlusion of atrial septal defects (ASDs) (Figure 4 and Figure 5) [12,13,14] and patent ductus arteriosus (PDA) (Figure 6) [15]. Similar actuarial analysis was performed to evaluate rates of resolution of residual shunts following the closure of ASDs [12,13,14] and PDAs [15,16,17].

### 3.3. Actuarial Analysis of Event Free Rates Using the Grunkemeier and Starr Method

A slightly different method of actuarial analysis (the method of Grunkemeier and Starr [18,19] was used for expressing patient survival and complication-free survival rates (Figure 7) as well as valve survival rates (Figure 8 and Figure 9) following prosthetic valve replacements in children [19,20,21]. The survival curves were compared using log-rank tests [20,21,22]. Thromboembolic and bleeding complications during anti-coagulant therapy following prosthetic valve replacements in children were expressed as linearized rates, i.e., the number of events per 100 patient-years, expressed as a percentage [20].

### 3.4. Linear Regression and Correlation Coefficients

Linear regression analysis was performed by comparing measured and calculated variables, and between different types of variables, depending on the type of study. Correlation coefficients were calculated to determine the statistical relationship between two variables; Spearman’s correlation coefficient (R) was used most often. This was generally thought to be better than the Pearson correlation coefficient by most statisticians that the author consulted with. Values closer to +1 or −1 were considered significant. An example is shown in Figure 10 in which left ventricular muscle mass was plotted against hemoglobin values with an R value of 0.74 and with a *p* < 0.001 [23]. Some investigators use R^2^ instead of R. We believe that R is acceptable and adequate in our studies. Other examples of correlations of Doppler and echo variables are shown in Figure 11, Figure 12, Figure 13, Figure 14 and Figure 15 [24,25,26].

### 3.5. Simple and Multiple Linear Regression Analysis

Simple and multiple linear regression was used to assess the relationship between independent variables, such as age, catheter size, and various indices of arterial pulse and limb growth [27]. Contingency tables were prepared at arbitrary cut-off points to discern differences in arterial insufficiency and limb growth [27]. A similar analysis with contingency tabulation of balloon inflation characteristics was helpful in the evaluation of the effectiveness of balloon pulmonary valvuloplasty [28,29]. Multiple linear regression was also used to assess the relationship between the independent main and interactive effects of stretch and recoarctation status and outcomes of gain and recoil [30].

### 3.6. Inter-Observer and Intra-Observer Variability

Inter-observer variability (measurements made by two or more observers examining the same data) and intra-observer variability (measurements made by the same observer at different times) were evaluated; a low percentage of variation assures the validity of the observations.

## 4. Aortic Remodeling Following Balloon Angioplasty of Aortic Coarctation

When we examined aortic remodeling following balloon angioplasty of aortic coarctation, we had to employ special methods to assess this issue.^31^ Inter- and intra-group comparisons of the shape of the aorta were needed for this process. Initially, measurements of the ascending aorta, isthmus, coarcted segment, and descending aorta immediately distal to the site of coarctation and at the level of the diaphragm (Figure 16) were made before angioplasty and at follow-up. For this assessment, it seemed appropriate to standardize the measurements to create a pure measure of shape. This was done by dividing each of the five aortic measurements by their average for each subject, before and after treatment. The resulting standardized measures would be on the same scale for each subject at each time point (Figure 17).

Once the standardized aortic measures were determined, the variance (the sum of the squared differences of each measure divided by the degrees of freedom) from norm or unity was determined for each patient before balloon angioplasty and at the follow-up study. Inter- and intra-group comparisons were made by using the variance of five standardized diameter values. This method was designed to assess how much the aorta deviated from the norm (an aorta with a uniform diameter throughout). The test statistic for the intergroup comparison was the nonparametric Mann-Whitney U statistic. To test for the effect of balloon angioplasty within each group, Wilcoxon’s signed rank test was used. The data (Figure 18) suggest that group A (good results) and group B (poor results) were not statistically different (*p* > 0.05) before angioplasty, though the variability of the aortic diameters in group B tended to be greater. At the follow-up study, the aortic measurements from group A patients were significantly more uniform (*p* = 0.01), suggesting a better remodeling in these patients with good results. There was improvement within both groups after angioplasty (*p* < 0.05); there was a greater percentage improvement (0.233 versus 0.057) in the group A patients than in the group B patients (0.287 versus 0.129). Cluster analysis was used to determine whether patients with good results could be differentiated from those with poor results on the basis of the shape of the aorta. Using Anderberg’s “nearest centroid sorting” method with the variance of the standardized diameter values as the defining characteristic, the subjects were sorted into two clusters before and after treatment. In examining the 2 × 2 tables of clusters and groups, it is apparent that cluster analysis does not distinguish between groups A and B before angioplasty (meaning that both groups are similar). However, with follow-up study they were found to be different.

Inspection of schematic diagrams of the aorta (Figure 19) graphically indicates that there was a definite remodeling of the aorta in group A (with good intermediate-term follow-up results), whereas there was less remodeling in group B with poor and fair results (Figure 20) [31].

Our study indicated that a better remodeling of the aorta takes place following successful balloon angioplasty of coarctation of the aorta, suggesting that normalized blood flow across dilated coarctation site encourages optimal growth of the aorta.

## 5. Summary and Conclusions

The statistical treatment of study materials used in multiple studies conducted by the author were reviewed. First, the data were scrutinized to ensure normal distribution, and expressed as mean ± standard deviation (SD) or standard error of mean (SEM) for normally distributed variables, while medians and ranges were given for data with skewed distribution. Two tailed, paired *t* tests or independent sample *t* tests (analysis of variance) were used for normally distributed data and non-parametric chi-square or similar other tests for data with skewed distribution, as appropriate. A value of *p* < 0.05 was set for statistical significance and Bonferroni correction was applied when multiple comparisons were made. A number of other types of statistical methods, including multivariate logistic regression, actuarial analysis of event-free rates, linear regression, correlation coefficients, simple and multiple linear regression analysis, contingency tables, and inter-observer and intra-observer variability were utilized as deemed appropriate. In the final section, special methods used in assessing aortic remodeling following balloon angioplasty of native aortic coarctation were detailed.

## Figures and Tables

**Figure 1 children-08-00296-f001:**
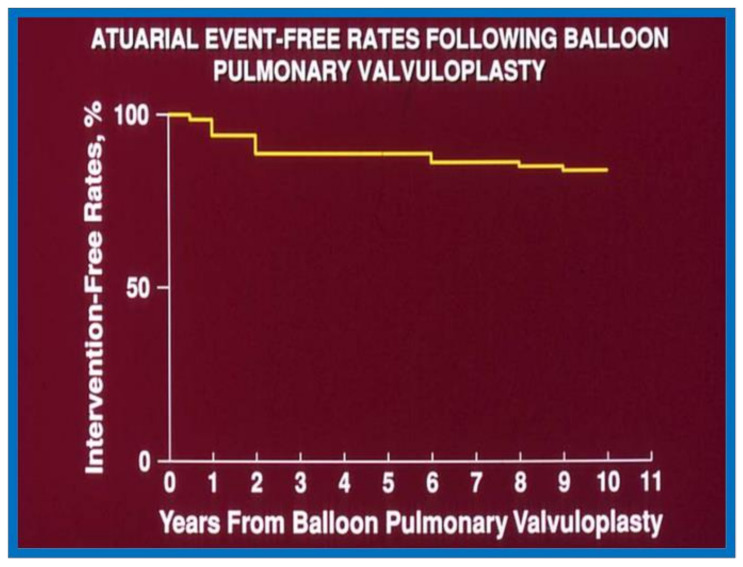
Actuarial event-free rates after balloon pulmonary valvuloplasty. Re-intervention-free rates at one, two, five, and 10 years after the procedure are 94%, 89%, 88%, and 84%, respectively. Reproduced from reference [9].

**Figure 2 children-08-00296-f002:**
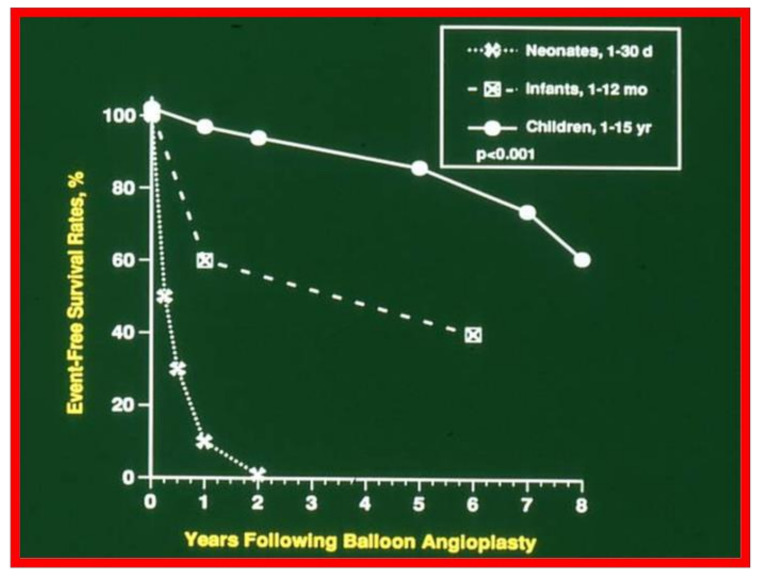
Actuarial event-free survival curves of neonates (<30 days), infants (1 to 12 months) and children (1 to 15 years) who had undergone balloon angioplasty of aortic coarctation. The event-free survival rates are better for the children than for the neonatal and infant groups (*p* < 0.001). Reproduced from reference [10].

**Figure 3 children-08-00296-f003:**
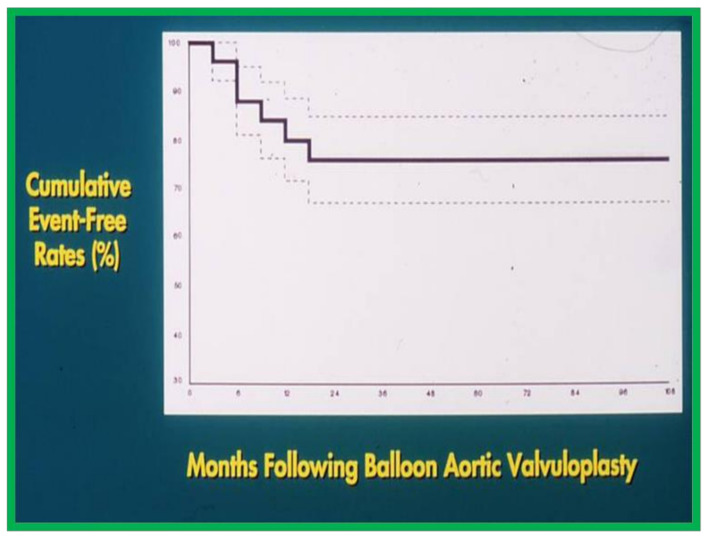
Actuarial event-free rates after balloon aortic valvuloplasty. 70% confidence limits are marked with dashed lines. Note intervention-free rates at 1, 2, 5, and 9 years are 80%, 76%, 76%, and 76%, respectively. Modified from reference [11].

**Figure 4 children-08-00296-f004:**
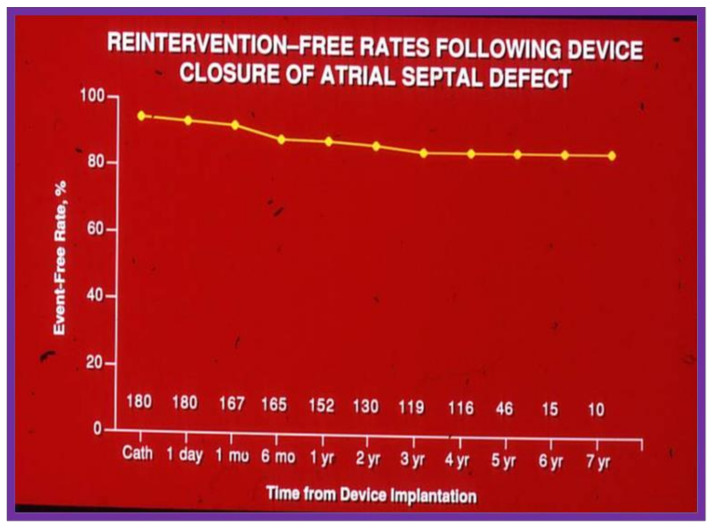
Graph depicting actuarial event-free rates following trans-catheter buttoned device occlusion of atrial septal defects. Note high (85%) event-free rates at 7 years following device implantation. Modified from reference [14].

**Figure 5 children-08-00296-f005:**
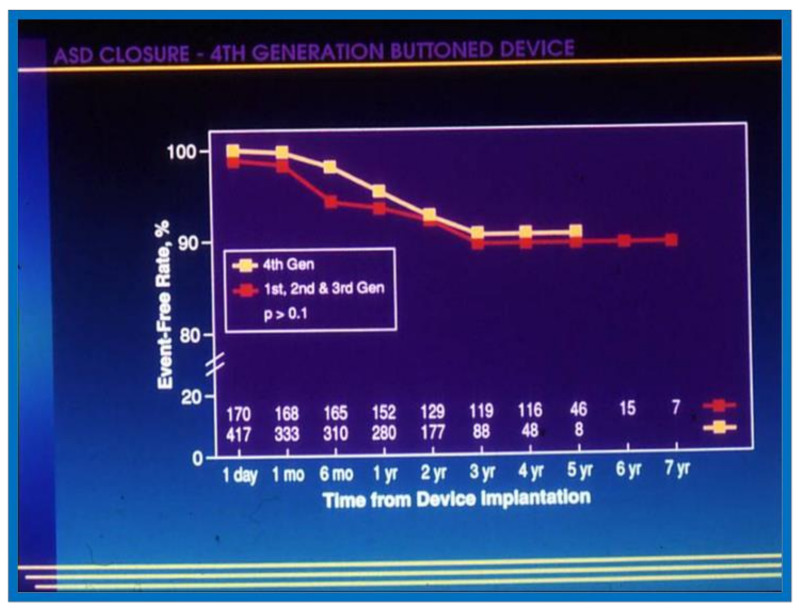
Graph comparing event-free rates after successful device implantation of first, second and third generation vs. fourth generation buttoned device to close secundum atrial septal defects. The fourth generation (Gen) data are depicted by filled yellow squares and the first, second and third generation by filled red squares. The number of patients available for follow-up at each specified follow-up interval is shown at the bottom of the graph. No difference (*p* > 0.1) by log-rank test was seen between the two cohorts. Modified from reference [13].

**Figure 6 children-08-00296-f006:**
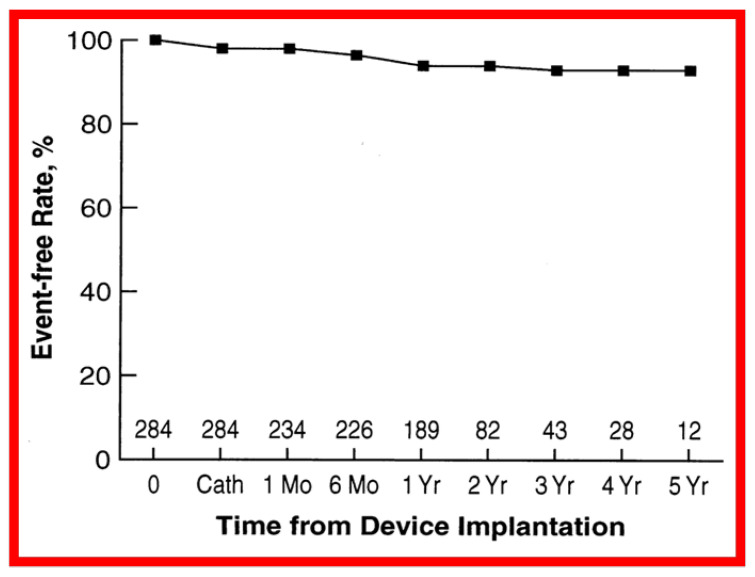
Graph showing actuarial event-free rates after transvenous buttoned device occlusion of patent ductus arteriosus. Reproduced from reference [15].

**Figure 7 children-08-00296-f007:**
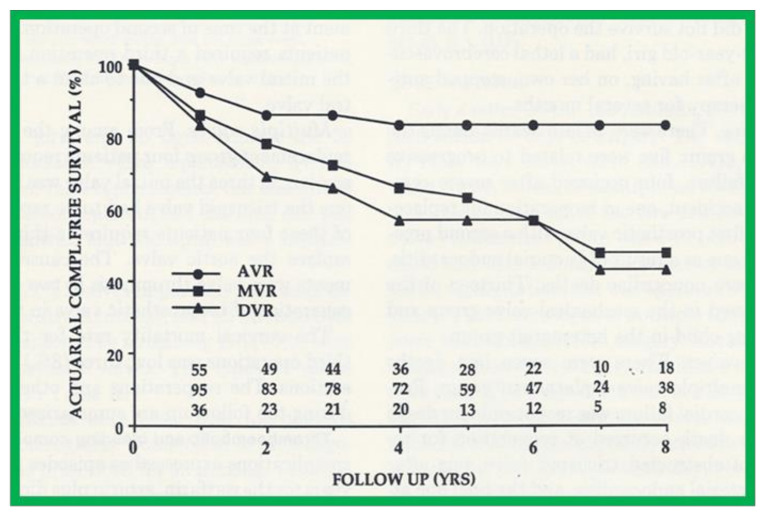
Actuarial major complication-free survival curves for aortic (AVR), mitral (MVR), and double (DVR) valve replacement are shown. Complication-free survival rates are better for AVR than for MVR and DVR. Similar complication-free survival rates for MVR and DVR suggest that mitral prostheses are largely responsible for complications. Reproduced from reference [21].

**Figure 8 children-08-00296-f008:**
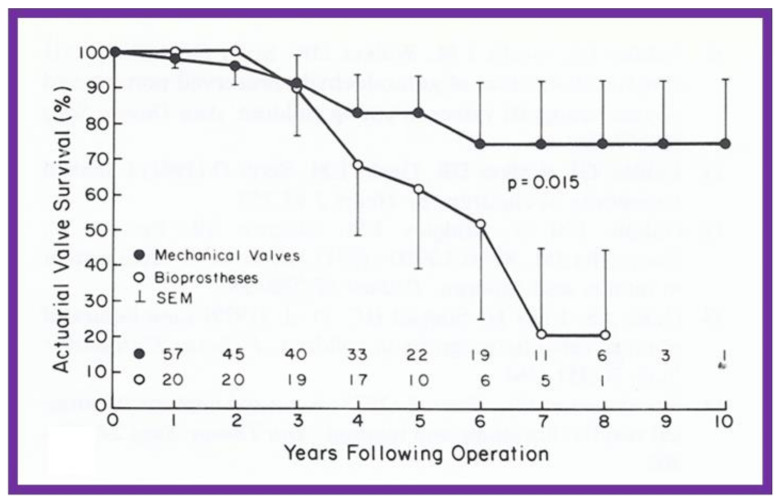
Actuarial valve survival curves for children ≤15 years are shown for mechanical (closed circles) and porcine heterografts (open circles); note the poor survival rate for heterografts (*p* = 0.015). The confidence limits are marked on only one side of the curve to clearly differentiate both curves. Reproduced from reference [22].

**Figure 9 children-08-00296-f009:**
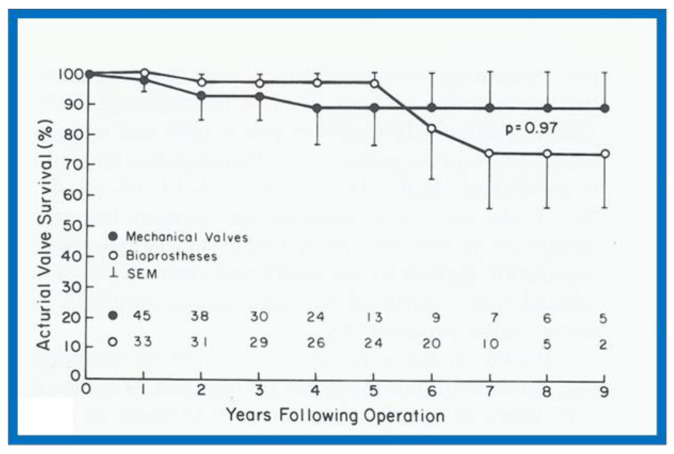
Actuarial valve survival curves for children > 15 years are shown for mechanical and porcine heterografts; the survival curves are similar (*p* = 0.97). The confidence limits are marked on only one side of the curve in order to visualize both curves clearly. Reproduced from reference [22].

**Figure 10 children-08-00296-f010:**
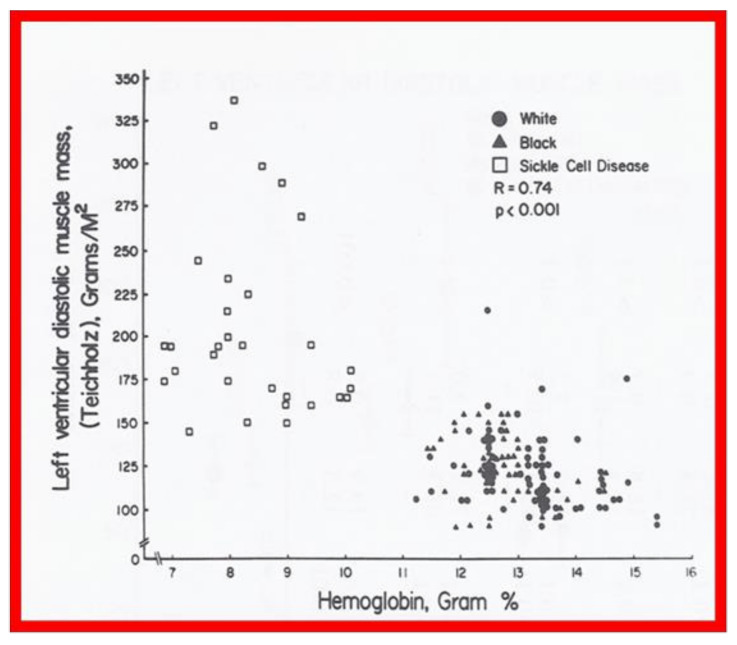
Left ventricular muscle mass in diastole, calculated using the Teichholz method, is plotted against hemoglobin. Note the significant (R = 0.74; *p* < 0.001) correlation between these parameters. Similar correlations were noted between the diastolic and systolic left ventricular muscle mass, calculated by all three methods on the one hand, and hemoglobin values on the other. Reproduced from reference [23].

**Figure 11 children-08-00296-f011:**
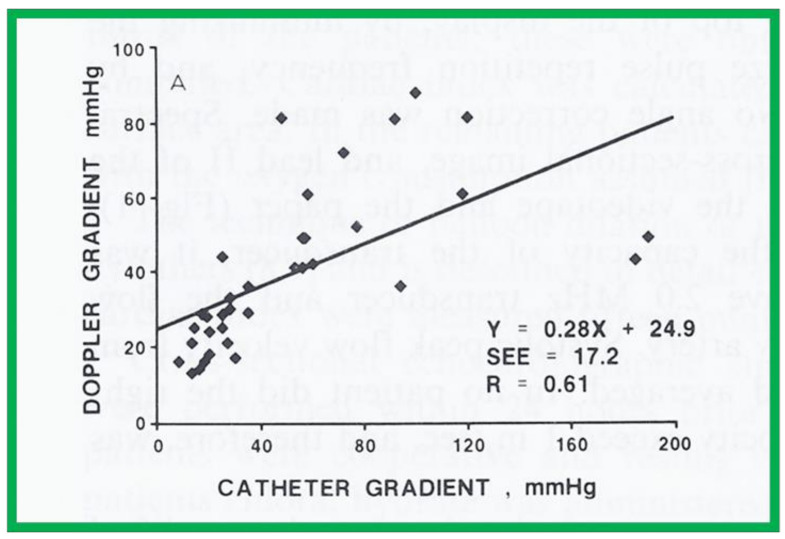
A scattergram demonstrating the relationship of Doppler-derived (by modified Bernoulli equation) peak instantaneous and catheterization-measured peak-to-peak pulmonary valve systolic pressure gradients is shown. Note that the linear regression analysis indicated a correlation coefficient (R) of 0.61. Reproduced from reference [24] Rao PS. International J Cardiol 1987; 15:195-203.

**Figure 12 children-08-00296-f012:**
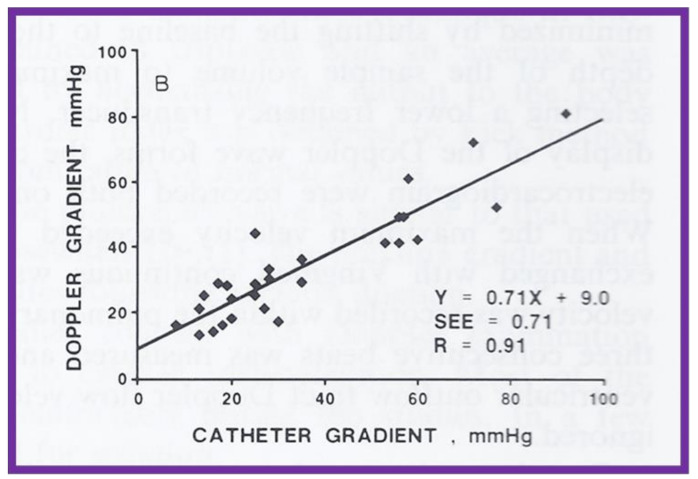
A scattergram demonstrating the relationship of Doppler-derived (by modified Bernoulli equation) peak instantaneous and catheterization-measured peak-to-peak pulmonary valve systolic pressure gradients is shown; this is similar to Figure 11, but after the removal of data sets from five patients with severe stenosis and one patient with severe infundibular stenosis. Note that the linear regression analysis indicated improvement in the correlation coefficient (R) to 0.91. Reproduced from reference [24].

**Figure 13 children-08-00296-f013:**
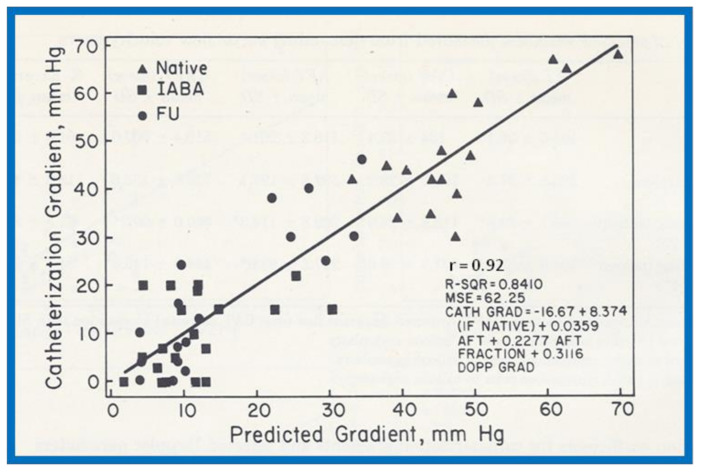
Linear regression analysis of catheterization-measured peak-to-peak gradients across aortic coarctation and predicted gradient calculated by the formula shown in the text indicated a better correlation (r = 0.92). Filled triangles: native coarctations; filled squares: coarctations immediately after balloon angioplasty (IABA); filled circles: coarctations at follow-up (FU). Reproduced from reference [25].

**Figure 14 children-08-00296-f014:**
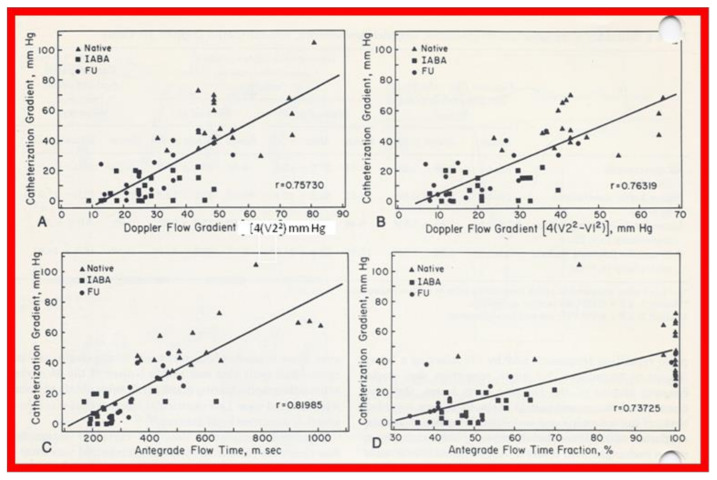
Linear regression analysis of catheterization-measured peak-to-peak and Doppler-derived (modified Bernoulli equation) peak instantaneous gradients across aortic coarctation are shown in (**A**,**B**). Note the similar correlation coefficients irrespective of the inclusion of proximal Doppler velocities. Similar regression analysis of catheterization-measured peak-to-peak gradients and antegrade flow time (milli seconds) (**C**) and antegrade flow time fraction (%) (**D**) shows minimal increase in correlation coefficient (r = 0.82) when antegrade flow time is used. Filled triangles: native coarctations; filled squares: coarctations immediately after balloon angioplasty (IABA); filled circles, coarctations at follow-up (FU). Reproduced from reference [25].

**Figure 15 children-08-00296-f015:**
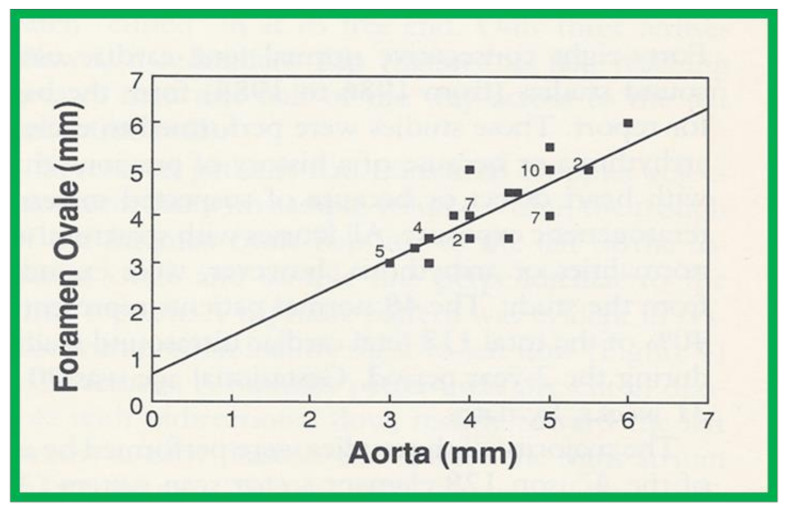
Plot of the diameter of the foramen ovale against the diameter of the aorta. The numbers indicate the number of subjects with that particular measurement. Note the excellent correlation with an r value of 0.84, y intercept of 0.605 and slope of 0.817. Reproduced from reference [26].

**Figure 16 children-08-00296-f016:**
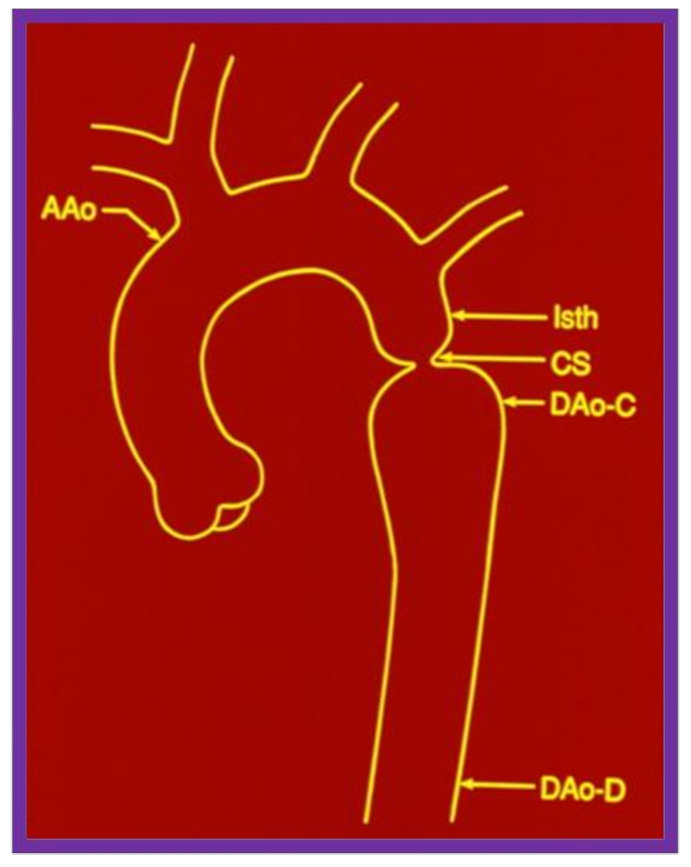
The diagram shows measurements of the aorta at five sites: the ascending aorta proximal to the origin of the right innominate artery (AAo); the isthmus (Isth); the coarcted aortic segment (CS); the descending aorta distal to the coarctation (DAo-C); and at the level of the diaphragm (DAo-D). These measurements were made on the angiograms performed prior to balloon angioplasty and at follow-up, to determine the extent to which remodeling of the aorta had occurred. The measurements were made in two angiographic views, corrected for magnification and averaged. Modified from reference [31].

**Figure 17 children-08-00296-f017:**
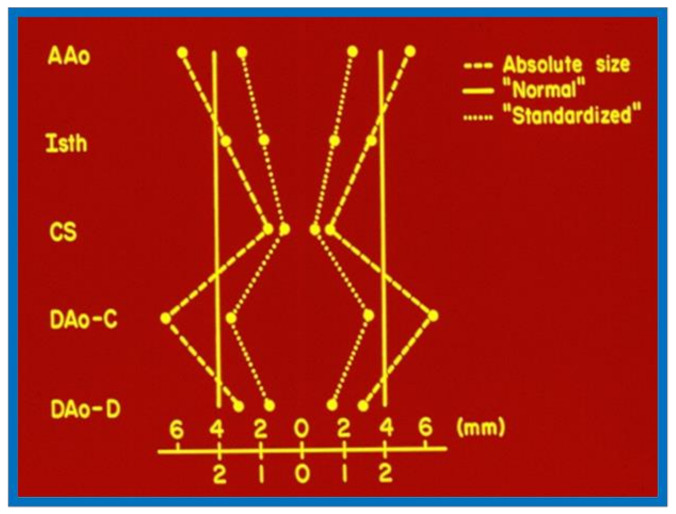
The diagram shows how the standardized diameters of the aorta at the five locations were calculated for each case before angioplasty and at follow-up study. The absolute sizes (dashed line) at each of the five locations were averaged; the averages are represented by solid lines. The standardized aortic measurement of each site is calculated by dividing the absolute size by the average of all five measurements. The dotted line represents the aortic shape, which can be compared with that of other patients and after intervention. Abbreviations are the same as those used in Figure 16. Modified from reference [31].

**Figure 18 children-08-00296-f018:**
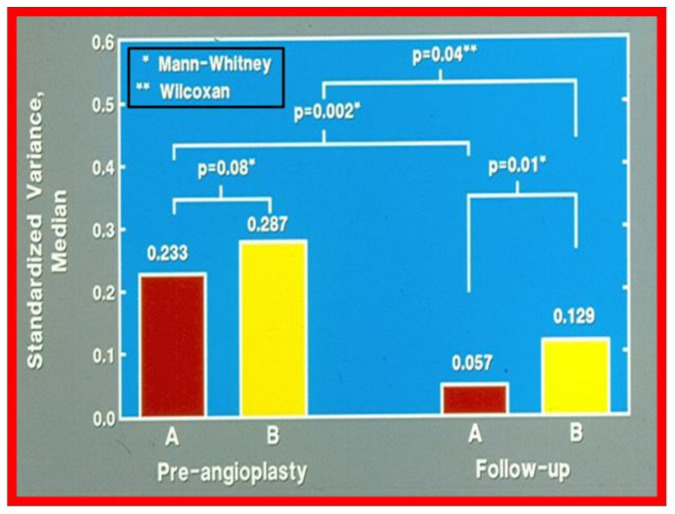
Bar graph showing comparison of the variances of standardized aortic diameters between groups A (good results) and B (poor results). The variance was similar (0.233 vs. 0.287; *p* > 0.05) in both groups before angioplasty. However, at follow-up the variances were different (0.057 vs. 0.129; *p* = 0.01). There was also a greater percentage improvement at follow-up study (0.233 vs. 0.057; *p* = 0.002) in group A, which had good results, than in group B which had fair or poor results (0.287 vs. 0.129; *p* = 0.04). The type of nonparametric test used for comparison is denoted in the insert at the left upper corner. Reproduced from reference [1].

**Figure 19 children-08-00296-f019:**
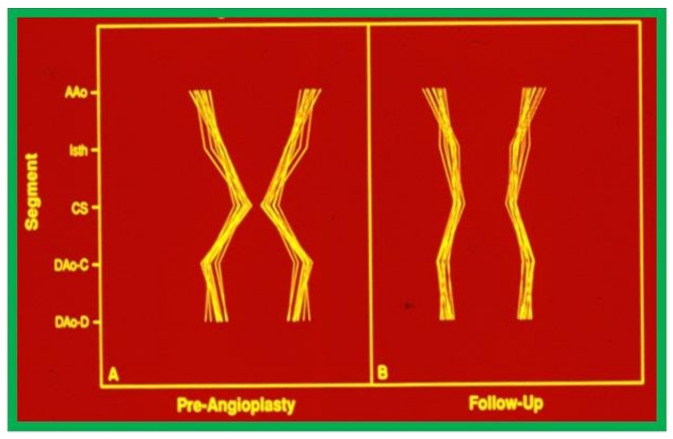
Schematic diagram of standardized aortic diameters pre-angioplasty (**A**) and at follow-up (**B**) in group A, which had good results. Note the improvement in that there is more uniformity of the various diameters of the aorta. Abbreviations are same as those used in Figure 2, Figure 3, Figure 4, Figure 5, Figure 6, Figure 7, Figure 8, Figure 9, Figure 10, Figure 11, Figure 12, Figure 13, Figure 14, Figure 15, Figure 16 and Figure 17. Modified from reference [31].

**Figure 20 children-08-00296-f020:**
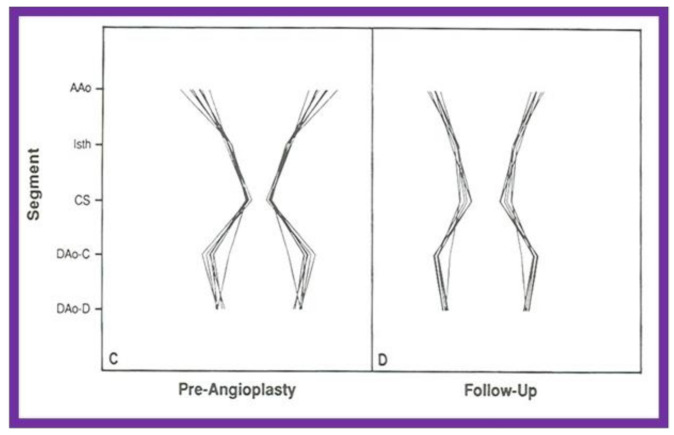
Schematic diagram of standardized aortic diameters pre-angioplasty (**C**) and at follow-up (**D**) in group B, which had poor results. Note that there is no significant improvement in the diameters of the aorta. Abbreviations are same as those used in Figure 16 and Figure 17. Reproduced from reference [31].

**Table 1 children-08-00296-t001:** Standard statistical analysis employed.

Parameter Examined	Methods Used
Are the data normally (Gaussian) distributed	Kolmogorov-Smirnov or similar tests
If normally distributed	Mean ± standard deviation (SD) or standard error of mean (SEM)
For data with skewed distribution	Medians and ranges
Comparison of pre vs. post procedure or intervention	Two-tailed or paired *t* tests
Between-group comparisons of categorical, ordinal, or not normally distributed variables	Fisher’s exact, Kruskal-Wallis, McNemars, or Mann-Whitney tests or other chi-squared tests
Level of statistical significance	*p* < 0.05
Multiple comparisons	Bonferroni correction

**Table 2 children-08-00296-t002:** Specialized statistical methods employed.

Multivariate Logistic Regression
Actuarial Analysis of Event Free Rates
Actuarial Analysis of Event Free Rates Using the Grunkemeier and Starr Method
Linear Regression and Correlation Coefficients
Simple and Multiple Linear Regression Analysis
Inter-Observer and Intra-Observer Variability

## Data Availability

Not applicable.

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
