# Peer review of "Statistical Treatment of Clinical Investigations in Pediatric Cardiology"

_children, 2021, doi:10.3390/children8040296_

Round 1
Reviewer 1 Report
This is a good review perspective titled: Statistical treatment of clinical investigations in pediatric cardiology written by Dr. P. Syamasundar Rao.
This paper is written with a goal of providing a broad perspective of statistical use in the field of pediatric cardiology. Dr. Rao has to be commended for the extensive and significant contribution provided. Several of them are landmark studies in the field of pediatric interventional cardiology.
It would be helpful as part of review if there is more description of the individual statistical methods utilized.
Author Response
Thanks for the complimentary comments of the reviewer. More detailed description of each statistical method is beyond the scope of the presentation. However, these statistical methods have been referenced to the appropriate paper.
Reviewer 2 Report
Excellent paper giving a perspective of use of statistical methods in pediatric cardiology papers published by the author given the author's long-standing experience in the field.
My comments:
- p value: why did the author use a pre fixed value of 0.05 in his studies. Recently, thee has been an uproar against the use of a fixed p value. For example: In this paper by O’Connor CM "A call for change: level of statistical significance." published in JACC it is very nicely explained that use of p value <0.05 can impact the way physicians understand clinical significance of the comparisons made.
- In the use of corelation coefficients, the author mentions that Spearman corelation was the most commonly used but fails to mention the reason for the same. It would be beneficial for the readers if the author was to rationalize the decision for using it more often than Pearson corelation
- When the author mentions multiple linear regression, only r is visualized in the figures. It is worth mentioning the utility of r 2 (coeffciient of determination) as compared to only using r (correlation coefficient). The coefficient of determination, r 2, is useful because it gives the proportion of the variance (fluctuation) of one variable that is predictable from the other variable. It is a measure that allows us to determine how certain one can be in making predictions from a certain model/graph.
Author Response
Thanks for the reviewer's complimentary remarks. Item 1. I agree with the reviewer and that of O'Conner regarding impact on use of p value <0.05. But my review is on the studies used in the past and can't now be changed. 2. Spearman correlation was often used statistical analysis for the type of respective materials in the study subjects. This was thought to be better than Pearson correlation by most statisticians that the author consulted with. A sentence is added to the discussion accordingly 3. With regard to use of r vs. r 2 , while I agree with the reviewer, we chose r instead of r 2 in our studies at that time. I have accordingly added couple of sentences in that section of the manuscript.